# Blue light-emitting diodes based on colloidal quantum dots with reduced surface-bulk coupling

Xingtong Chen[1,2], Xiongfeng Lin[3], Likuan Zhou[3], Xiaojuan Sun[1,2], Rui Li[1,2], Mengyu Chen[1,2], Yixing Yang[3], Wenjun Hou[3], Longjia Wu [iD][3] ✉, Weiran Cao[3], Xin Zhang[3], Xiaolin Yan[3] & Song Chen [iD][1,2] ✉

To industrialize printed full-color displays based on quantum-dot light-emitting diodes, one must explore the degradation mechanism and improve the operational stability of blue electroluminescence. Here, we report that although state-of-the-art blue quantum dots, with monotonically-graded core/shell/shell structures, feature near-unity photoluminescence quantum efficiency and efficient charge injection, the significant surface-bulk coupling at the quantum-dot level, revealed by the abnormal dipolar excited state, magnifies the impact of surface localized charges and limits operational lifetimes. Inspired by this, we propose blue quantum dots with a large core and an intermediate shell featuring nonmonotonically-graded energy levels. This strategy significantly reduces surface-bulk coupling and tunes emission wavelength without compromising charge injection. Using these quantum dots, we fabricate bottom-emitting devices with emission colors varying from near-Rec.2020-standard blue to sky blue. At an initial luminance of 1000 cd m$^{-2}$, these devices exhibit $T_{95}$ operational lifetimes ranging from 75 to 227 h, significantly surpassing the existing records.

Printed display technologies, based on solution-processable electroluminescent materials, feature low-cost and scalable manufacturing on flexible substrates. Among them, quantum-dot light-emitting diodes (QLEDs) are particularly interesting due to their premium display quality[1–6]. So far, QLEDs' electroluminescence (EL) quantum efficiencies have reached the theoretical maximum[7–9]. The $T_{95}$ operational lifetime, defined as the time for luminescence to degrade to 95% of the initial value ($L_0$), has exceeded 5000 h at $L_0 = 1000$ cd m$^{-2}$ for red and green devices[9–11]. However, as a general challenge faced by solution-processable emitters, the electroluminescence stability of blue QDs has not met the industrialization standard. The best results for sky blue and near-Rec.2020-standard blue is 57 and 30 h, respectively[6,11].

The enhancement of QLED lifetimes relies on understanding the degradation mechanism, for which the discussion was limited to the hole-transporting layer (HTL) and its coupling with QD emitters[12]. However, the space charge effect or the hot electron effect, causing the slow degradation of red QLEDs, cannot explain the fast degradation of blue devices[13]. In comparison, the operation-induced QD degradation remains under-explored[14]. On the one hand, the monotonically-graded core/shell/shell structures are currently the best solution to balance electrical and optical performance[8,15]. On the other hand, our previous work indicates that these state-of-the-art blue QDs are intrinsically more susceptible to charge accumulation than their red counterparts[13]. This dilemma underlines the necessity of studying and innovating the QD structure.

This work first reveals that QDs with state-of-the-art monotonically-graded core/shell/shell structures and stable surface ligands show significant irreversible photoluminescence (PL) loss during QLED

[1]Suzhou Key Laboratory of Novel Semiconductor-optoelectronics Materials and Devices, College of Chemistry, Chemical Engineering and Materials Science, Soochow University, Suzhou 215123 Jiangsu, China. [2]Jiangsu Key Laboratory of Advanced Negative Carbon Technologies, Soochow University, Suzhou 215123 Jiangsu, China. [3]TCL Corporate Research, Shenzhen 518067 Guangdong, China. ✉e-mail: wulongjia@tcl.com; songchen@suda.edu.cn

operation. The spectral analysis uncovers that the abnormal dipolar characteristics magnify the impact of surface localized charges at the QD level, limiting the operational lifetimes of blue QLEDs. Next, we design QDs with a large core and nonmonotonically-graded shells to reduce surface-bulk coupling. Finally, record-breaking $T_{95}$ lifetimes are achieved for blue QLEDs at different emission wavelengths, corresponding to chromaticity values from CIE-y = 0.063–0.177. Under the premise of efficient charge transport, the strong correlation between surface-bulk coupling and electroluminescence stability is pronounced.

## Results and discussion

### Charge-induced degradation of state-of-the-art blue QDs

Currently, the most successful blue QDs apply the monotonically-graded core/shell/shell structure like Q1[9,11,14]. With a composition of $Zn_{0.8}Cd_{0.2}Se/Zn_{0.7}Cd_{0.3}Se_{0.4}S_{0.6}/ZnS$ (see Fig. 1a and Supplementary Figs. 1 and 2), each dot is capped with octanethiol, which is electrochemically stable according to a recent study[14]. Compared with our previous reports[8,13], Q1 features a finely-tuned shell composition and enhanced carrier injection (Supplementary Figs. 3 and 4a). Therefore,

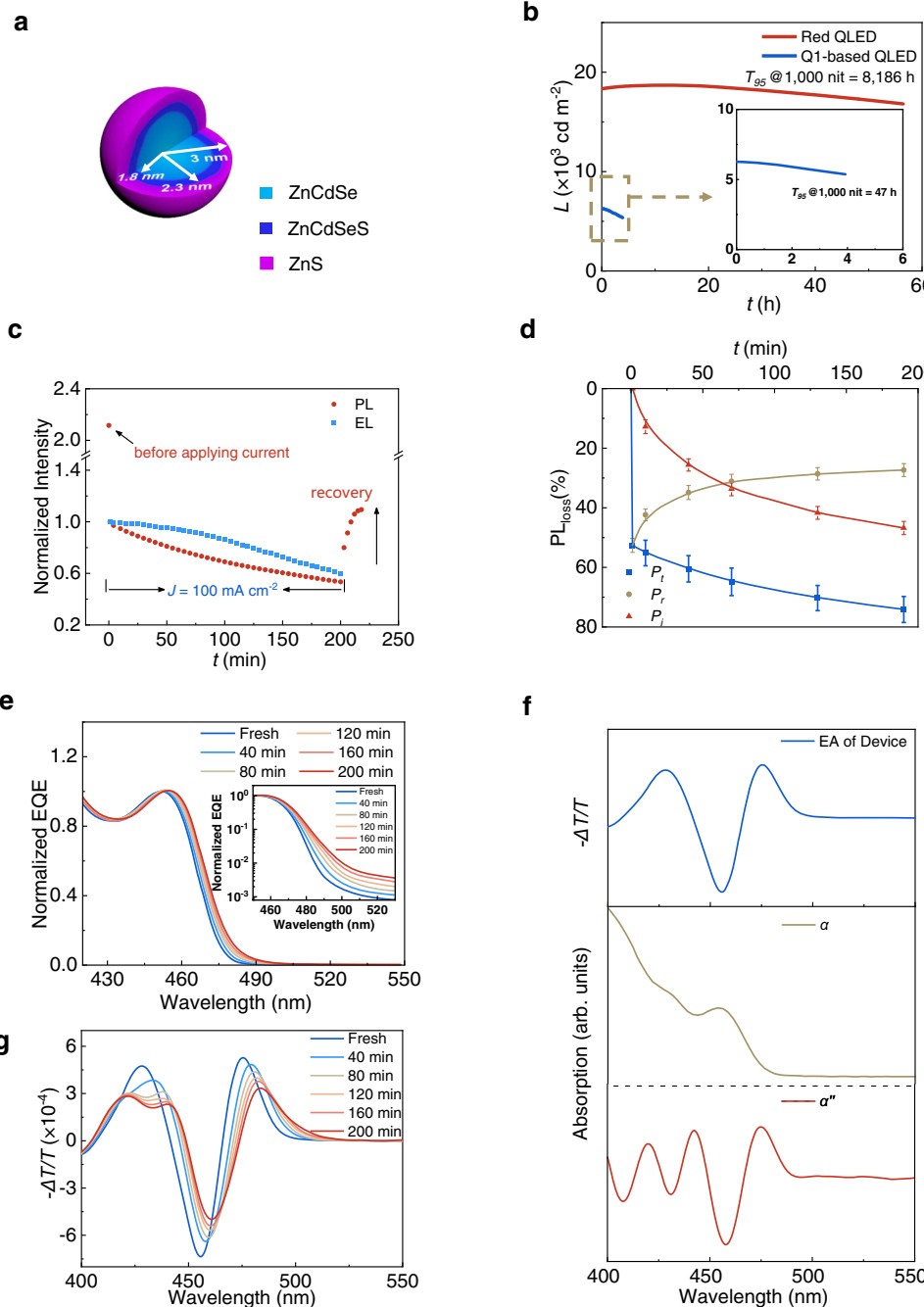

**Fig. 1 | Charge-induced degradation in state-of-the-art blue QDs. a** The core/ shell/shell structure of Q1 ($Zn_{0.8}Cd_{0.2}Se/Zn_{0.7}Cd_{0.3}Se_{0.4}S_{0.6}/ZnS$). The radial element distribution is obtained by measuring high-resolution TEM and ICP-OES. **b** Operational lifetime of red and Q1-based devices. The current density is set at 150 and 50 mA cm⁻², respectively. **c** PL and EL intensity of Q1-based QLEDs simultaneously monitored as a function of operation time. **d** Reversible, irreversible, and total PL loss of Q1 measured from corresponding QLEDs as a function of device operation time. The current density is set at 100 mA cm⁻². The error bars represent the data range. **e** Sensitive external quantum efficiency (sEQE_PV) spectra of Q1-based QLEDs. The inset shows the same sEQE_PV plotted on a logarithmic scale to demonstrate the near band-edge transitions. **f** Upper: electroabsorption spectra of Q1-based QLED. Below: Q1's absorption coefficient ($\alpha$) and 2nd derivative of $\alpha$ ($\alpha''$). **g** Time-dependent electroabsorption spectra of Q1-based QLEDs. The devices for time-dependent spectral analysis are aged by a constant current density of 100 mA cm⁻².

the Q1-based QLED exhibits external quantum efficiency (EQE) of 19.2% and a $T_{95}$ lifetime ($L_0 = 1,000$ cd m$^{-2}$, $\lambda = 474$ nm) of 47 h (see Fig. 1b and Supplementary Fig. 5b), close to the best results reported so far despite shorter emission wavelength[11,14]. To study the charge-induced degradation, we simultaneously monitor the EL and PL as a function of device operation time[7,16]. As seen in Fig. 1c, Q1's EL and PL decay synchronously after the steep PL drop at the beginning. Here, $(I_0-I_t)/I_0$ defines the total PL loss ($P_t$), where $I_t$ is the real-time PL photon flux, and $I_0$ is the initial value before applying current. Supplementary Fig. 6 also show the results measured from red QDs, which are ultrastable in PL despite the monotonically graded core/shell/shell structure.

If one interrupts the EL-PL test and applies continuous pulses of reverse bias to the device, the PL intensity recovers in the fashion shown in Fig. 1c, d. After saturation, the remaining PL loss is QDs' irreversible degradation ($P_i$), and the recovered part ($P_r$) is attributed to temporary charging. The sum of $P_i$ and $P_r$ equals $P_t$ (see Supplementary Fig. 7a, b). By recording the values of $P_i$ and $P_r$ at different test stages (see Supplementary Fig. 7c, d), we have Fig. 1d and Supplementary Fig. 6b. Interestingly, the PL loss for both devices is nearly 100% recoverable ($P_i = 0$, $P_t = P_r$) if the reverse bias is applied immediately after the initial PL drop (see Supplementary Fig. 8). Since then, both $P_r$ and $P_i$ have remained stable for the red QLED. In contrast, both $P_i$ and the ratio of $P_i/(P_r + P_i)$ of the Q1-based device increase monotonically up to 46.7% and 63.1% after being tested for 190 min, respectively, suggesting a growing number of dark QDs. The $P_r$ value decreases monotonically after the initial surge, which means $P_r$ continuously transforms to $P_i$. Here, both types of QDs are intrinsically ultrastable (see Supplementary Fig. 9), and the hot electron damage to HTL should decrease with reduced electron injection[13]. Therefore, the charge-induced irreversible degradation of Q1 is responsible for the EL degradation.

The excitonic transition of Q1 is first studied by measuring the EQE of QLED in the photovoltaic mode (sEQE$_{PV}$) as it provides a better resolved excitonic signal (Fig. 1e) than plain absorption techniques (see Supplementary Fig. 10a). Upon device degradation, the first absorption peak of the QDs is redshifted by 22.6 meV, partially recovered by applying a reverse bias (see Supplementary Fig. 10b). The inset of Fig. 1e shows that the weakly absorptive tail to the red of the bright peak increases significantly[17].

We use electroabsorption (EA) to resolve the spectral change further. The upper panel of Fig. 1f shows the EA spectrum measured from a Q1-based QLED. The lineshape ranging from 440 to 500 nm resembles the second derivative of the QD's absorption curve (see the bottom panel). According to the established theories[18–20], the second derivative lineshape and the quadratic field dependence (see Supplementary Fig. 11) prove the existence of a dipole moment in Q1's first excited state[21]. In the particle-in-sphere model[22], the wavefunction of the first excited state is spherically symmetric. Therefore, the abnormal dipolar characteristics suggest optical excitation involving the QD surface despite surface passivation (PLQE = 93% in Supplementary Fig. 12)[23–25]. In contrast to Q1, the large-size red QDs show EA lineshape resembling the first derivative of their absorption curves[13], suggesting a nonpolar excited state (Supplementary Fig. 13a).

Besides revealing the surface-bulk coupling, EA can detect the QD degradation caused by device operation. Figure 1g shows the first excitonic peak redshifts by 33.6 meV after device operation for 200 min, which is consistent with the sEQE$_{PV}$ results. Moreover, at the energies of the long absorption tail, the continuously distributed EA signal redshifts by over 100 meV, partially recoverable by reverse bias (see Supplementary Fig. 14a). Such a phenomenon has not been reported in the literature and is consistent with the enhanced near-band-edge transitions in the spectra of sEQE$_{PV}$ (Fig. 1e). More interestingly, as seen in Supplementary Fig. 13b, the ultrastable red QLEDs show no such spectral change. The origin of near-band-edge transitions in Cd-based chalcogenide QDs has been extensively studied. Compared with the existing models, the near-band-edge EA signals reported here

cannot be explained either by a single-level surface state[26–28] or dark exciton[29–31]. Instead, the second derivative lineshape and the continuous redshift of the EA signals suggest the domination of quasi-continuously distributed, surface-associated, and optically dark transitions[32,33].

The correlation between the EA spectra and QD degradation can be established. In an operating QLED, each irreversibly degraded QD carries at least a fixed charge, likely occupying a surface localized state. For a small-sized blue QD with a monotonically graded core/shell/shell structure (Q1), the substantial surface-bulk coupling causes a redshift of the bright transition peak (Fig. 1e, g) despite surface passivation[34–36]. Furthermore, since near-band-edge transitions are generally surface-associated, they are expected to interact more substantially with the surface charge and generate a larger spectral redshift (Fig. 1g).

## Blue QDs featuring reduced surface-bulk coupling

The most thorough way to reduce surface-bulk coupling is to passivate surface defects. However, surface-associated transitions are intrinsic to Cd-based chalcogenide QDs despite perfect ligand passivation[33]. Alternatively, one can modify the core and intermediate shell. For example, computational results predict that increasing the core size while maintaining the shell parameters can keep the exciton wavefunction away from the surface[37]. Applying this strategy, we adopt Q1's core/shell/shell composition and increase the core size (see Fig. 2a).

The radial element distribution of Q2 and Q3, measured by inductively coupled plasma optical emission spectroscopy (ICP–OES), is detailed in Fig. 2d, Supplementary Fig. 15, and Supplementary Fig. 16a–d. Figure 2d also shows the radial dependent electronic levels estimated by linearly combining the parameters of isolated materials. Figure 2b shows that both Q2 and Q3 show near-unity PL quantum efficiency (PLQE). Besides, due to the increase in core size, the Q2 and Q3 show narrow PL peaks at 476 and 481 nm, respectively.

Despite the potential reduction in the surface-bulk coupling, the increase in core size redshifts the emission color to sky blue, limiting the application in high-color-gamut displays. Synthesizing a monotonically-graded shell with a more substantial energy barrier is expected to realize a deeper blue. However, that inevitably obstructs charge injection. To avoid the dilemma, we propose an intermediate shell whose energy levels vary nonmonotonically with radius (see Fig. 2a). Such a design is explored to couple with large cores.

Figure 2a and Supplementary Fig. 16e–h show Q4's radial composition measured by ICP–OES. Linearly combining the energy levels of isolated materials using the ICP–OES results generates the energetic profile shown in Fig. 2d. Compared to the monotonically-graded shells, as in Q1-Q3, our design alleviates the contradiction between exciton confinement and carrier injection. The wider-gap sub-shell composed of ZnSeS provides an extra barrier to shield the exciton from the surface localized charge and blueshift the emission peak. More critically, the potential well within the ZnCdS sub-shell reduces the barrier width for charge injection, without inducing noticeable lattice strain and nonradiative recombination (see Supplementary Figs. 17, 19a, b). Except for the intermediate shell, the large core, the ZnS-based outer shell, and surface ligands are the same as Q2. Additional characterizations of Q4's C/S/S/S structure can be found in Supplementary Figs. 18 and 19. Towards the Rec.2020 standard, we increase the Zn/Cd ratio when synthesizing Q5's core, Zn$_{0.9}$Cd$_{0.1}$Se, while keeping the shell the same as Q4. Figure 2c shows that the nonmonotonically-graded shell blueshifts the emission peak of Q4 and Q5 to 469 and 465 nm, respectively. Both materials show near-unity PLQE. The evolution from Q1 to Q5 is summarized in Fig. 2a.

To verify the effect of the new QD structure, we quantify the degree of surface-bulk coupling in Q1-Q5. Figure 2e and Supplementary Fig. 20 show the dipole moments ($\mu$) and the average electron-hole separation ($d_{e-h}$) extracted from the EA spectrum of each type of QD at the assembly level. Considering that $d_{e-h}$ linearly scales with the nanocrystal radius for bare QDs[23], we use $d_{e-h}/r_c$ to quantify the degree

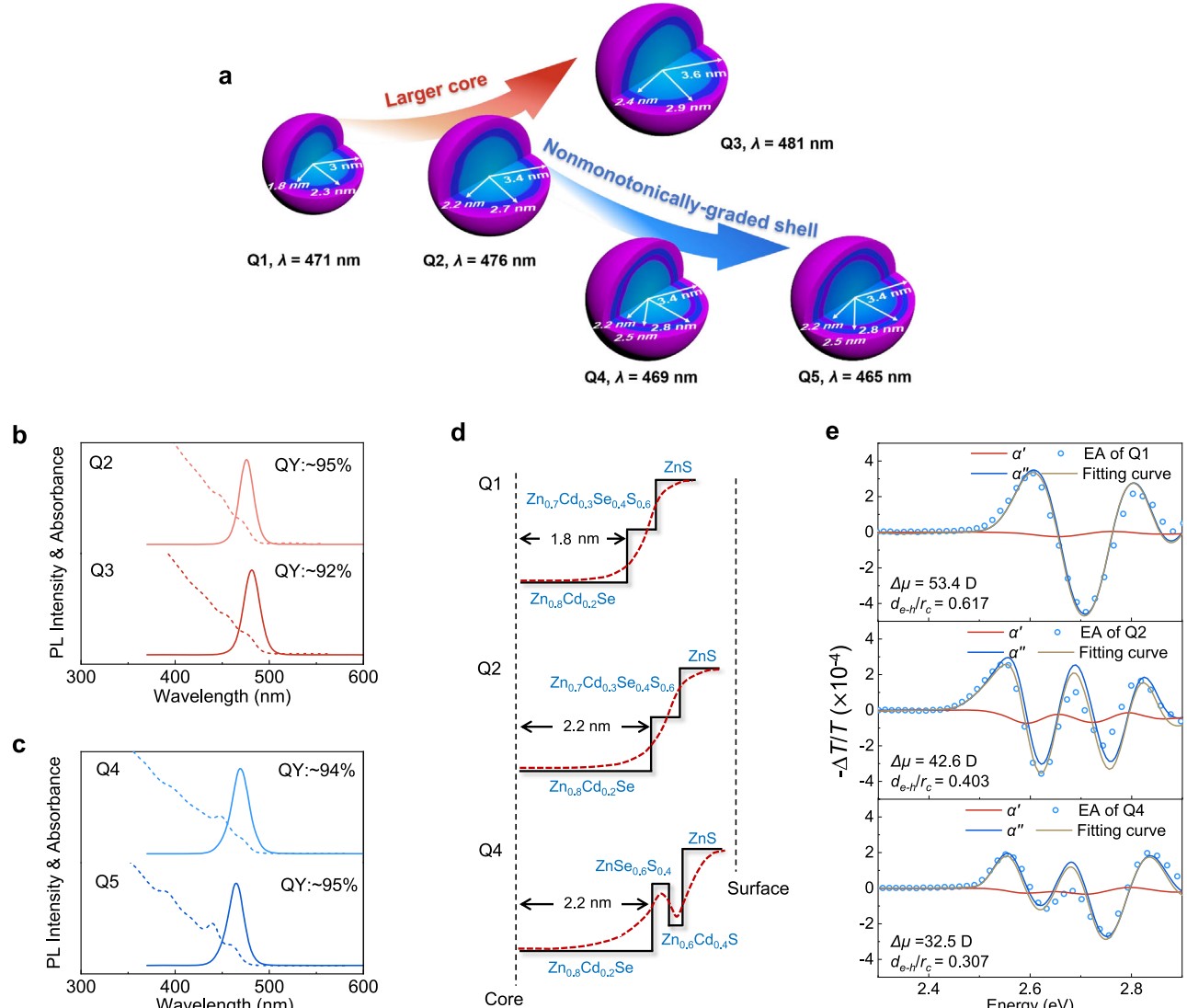

**Fig. 2 | Blue quantum dots with reduced surface-bulk coupling. a** The core/shell/shell structures for Q1-Q5. The radial element distribution is obtained using high-resolution TEM and ICP-OES. **b, c** PLQE, PL spectra, and absorbance spectra of Q2–Q5. **d** Schematic diagram of the conduction band minimum of Q1, Q2, and Q4. The electronic energy levels of the sub-shells are estimated by linearly combining the electronic energy levels of the isolated materials according to the measured radial compositional profiles. **e** Electroabsorption spectra and the extracted dipole moments of Q1, Q2, and Q4. The samples have a structure of ITO/PEDOT:PSS/QDs/Ag (-15 nm).

of surface-bulk coupling, in which $r_c$ is the core radius. Supplementary note 1 includes detailed methods. In Cd-based chalcogenide QDs, the energy levels make the electron more likely to couple with the surface than the hole[38,39]. Therefore, the $d_{e-h}$ value estimates the radial position of excited electrons averaged over the QD assembly. As seen in Fig. 2e, the dipole moment of Q1 (53.4 D) translates to a $d_{e-h}$ of 1.111 nm and a $d_{e-h}/r_c$ value of 0.617. In comparison, Q2 shows a smaller dipole moment of 42.6 D, a $d_{e-h}$ of 0.887 nm, and a $d_{e-h}/r_c$ value of 0.403. A further increase in core size makes Q3 even less dipolar ($\mu = 36.3$ D, $d_{e-h} = 0.756$ nm, $d_{e-h}/r_c = 0.314$). Figure 2e also shows the result of Q4. Despite the same core size as Q2, Q4's nonmonotonically-graded intermediate shell further reduces the $d_{e-h}/r_c$ value to 0.307. As expected, an increase in the bandgap of the core in Q5 increases the $d_{e-h}/r_c$ value to 0.361. As will be shown later, the difference in $d_{e-h}/r_c$ values plays a significant role in EL stability.

**Blue QLEDs with record-high operational lifetimes**

Bottom-emitting QLEDs are made using Q2-Q5. Figure 3a–c show the *L-J-V*, EQE, and operational lifetimes of bottom-emitting QLEDs based on

Q2 and Q3. Supplementary Figs. 4, 5, 21 summarize other relevant results. With the core size increased to 2.2 nm, the EL emission of Q2-based QLED peaks at 478 nm. The EQE reaches 19.7 %. The $T_{95}$ operational lifetime reaches 151 h ($L_0 = 1,000$ cd m$^{-2}$, $n = 1.80$), which is extrapolated to a $T_{50}$ ($L_0 = 100$ cd m$^{-2}$) of 50,206 h. With the core size further increased to 2.4 nm, the EL emission of Q3-based QLED peaks at 482 nm. The EQE reaches 20.4 %, and the operational lifetime is further improved to $T_{95} = 227$ h ($L_0 = 1000$ cd m$^{-2}$, $n = 1.79$), which is extrapolated to $T_{50} = 80,377$ h at $L_0 = 100$ cd m$^{-2}$. We summarize the operational stability in Fig. 3e (by luminance) and 3 f (by EQE). Compared with the most stable blue QLEDs reported in the literature[13,18], Q2- and Q3-based devices show similar CIE-y values but over three times the operational lifetimes (see Fig. 3e).

Figure 3a, b also show the *L-J-V*, EQE, and operational lifetimes of bottom-emitting QLEDs based on Q4 and Q5. Supplementary Figs. 4, 5, and 21 summarize other relevant results. With the core size of Q2 and the nonmonotonically-graded shell shown in Fig. 2a, Q4-based QLEDs generate EL emission centering at 471 nm. *L-J-V* characteristics confirm that the nonmonotonically-graded shell successfully blueshifts the

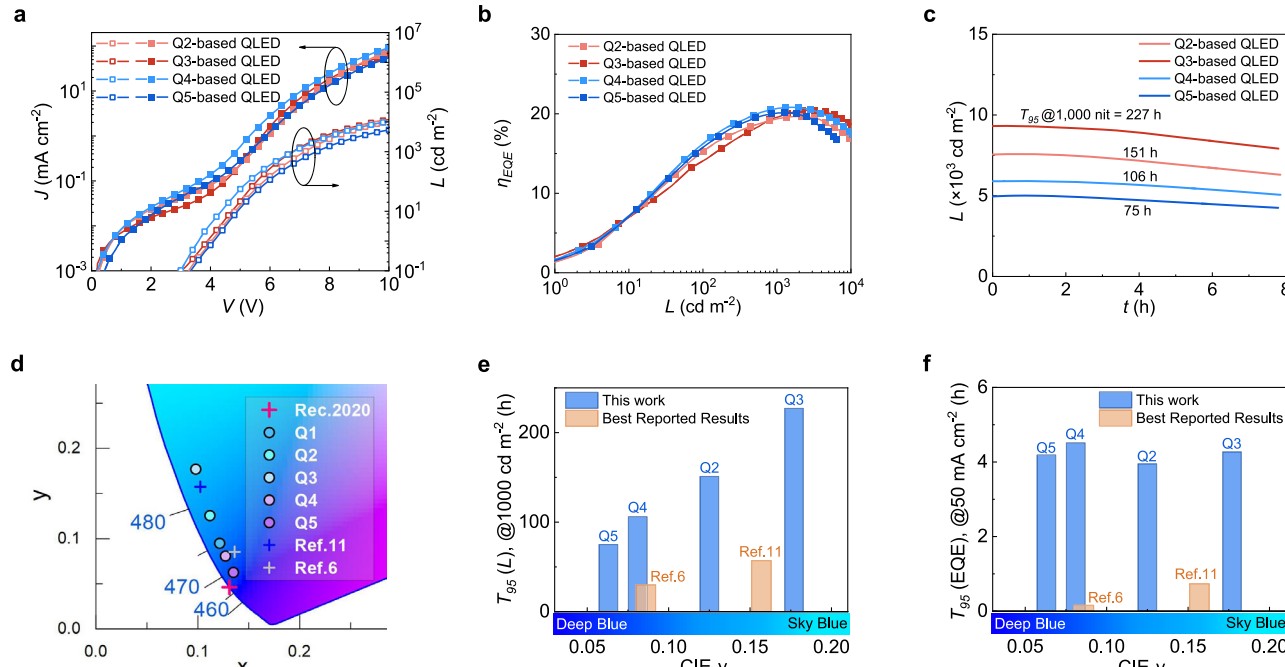

**Fig. 3 | Blue QLEDs with record-high operational lifetimes. a** $L$-$J$-$V$ characteristics. **b** External quantum efficiencies (EQEs). **c** Luminance decay of blue QLEDs based on Q2, Q3, Q4, and Q5. The current density is set at 50 mA cm$^{-2}$. **d** The CIE-1931 coordinates of Q1-, Q2-, Q3-, Q4, and Q5-based QLEDs (circles filled with different colors). The coordinates of the best-reported sky blue (ref. [11], the blue cross symbol), the best-reported near-Rec.2020-standard blue QLEDs (ref. [6], the gray cross symbol), and the Rec.2020 standard blue (the red cross symbol) are also marked. The reference data is extracted from the published graphs. **e** Histogram

summarizing the $T_{95}$ lifetimes defined as the time for luminescence to degrade to 95% of the fixed initial value (1000 cd m$^{-2}$). **f** Histogram summarizing $T_{95}$ lifetimes defined as the time for EQE to degrade to 95% of the initial value under a fixed current density (50 mA cm$^{-2}$). EQE lifetime reflects EL stability by excluding the effect of a human-eye response. The CIE-y values of each QLED are provided along with the color bar to highlight the difference in chromaticity. The reference data is extracted from the published graphs.

emission peak without compromising charge injection. The EQE reaches 20.8 %. As seen in Fig. 3c, the $T_{95}$ lifetime is 106 h ($L_0 = 1,000$ cd m$^{-2}$, $n = 1.78$). Notably, compared with the milestone work by Jang's group[6], the Q4-based QLED exhibits a CIE coordinate (0.127, 0.081) closer to the Rec.2020 standard and a $T_{95}$ lifetime 3.5 times longer (see Fig. 3e). Further, with an emission peak at 467 nm and a full width at half maximum of 20 nm, the CIE coordinate of Q5-based QLED locates at (0.135, 0.063), covering 97.4% Rec.2020 color gamut together with the Rec.2020 standard red and green. Despite the reduced eye sensitivity to deep blue, the $T_{95}$ ($L_0 = 1000$ cd m$^{-2}$) lifetime of Q5-based QLED still reaches 75 h, exceeding any previously published blue QLED. As seen in Fig. 3e and Table 1, devices based on Q2-Q5 show distinct advantages compared with any other reported near-Rec.2020-standard blue QLED[6,13,40]. Figure 3f summarizes the $T_{95}$ of EQE measured under the same current density, which removes the wavelength dependence of eye sensitivity functions. Under this measure, Q2-Q5's leading edge is further expanded. Specifically, Q4 is nearly 30 times more stable than the benchmarking device with a similar chromaticity[6]. To be mentioned below, the difference in $T_{95}$ (EQE) between Q2-Q5 is attributed to surface-bulk coupling. All the devices presented above show uniform emission (see Supplementary Fig. 22), which avoids the overestimation of $L_0$ and operational lifetime.

To confirm the mechanism of lifetime enhancement, we conduct EL-PL, sEQE$_{PV}$, and EA measurements. Figure 4 shows the results of Q2- and Q4-based devices. Those of Q3 and Q5 are included in Supplementary Figs. 7 and 23. After an operation at 100 mA cm$^{-2}$ for 190 min, the $P_i$ value of Q1 (Fig. 1e), Q2, and Q3 is 46.7%, 34.6%, and 23.1%, respectively. Meanwhile, $P_i + P_r$ and $P_i/(P_i + P_r)$ values follow the same trend. With the nonmonotonically graded shell, the $P_i$ values of Q4 (Fig. 4d) and Q5 (Supplementary Fig. 23) reduce to 20.5% and 24.6%,

respectively, suggesting slower charge-induced QD degradation. As further seen in Fig. 4c, f and Supplementary Fig. 23c, f, the first excitonic peaks of Q2, Q3, Q4, and Q5 redshift by 11.1, 10.8, 6.7, and 7.3 meV, respectively, suggesting that the large core and nonmonotonically-graded shell can mitigate the impact of surface charges on the first bright transition. Moreover, the redshifts of near-band-edge EA signals of Q2-Q5 are significantly lower than that of Q1, suggesting a reduced impact of surface localized charges on surface-associated dark transitions. Supplementary Fig. 24 shows that the snapshots of EA spectra taken at $T_{90}$, $T_{70}$, $T_{50}$, and $T_{30}$ are nearly identical for the devices based on Q1, Q2, and Q3, despite quite different lifetime values. The same conclusion also applies to Q4 and Q5. The correspondence makes sense because the spectral change and the degradation of QDs are charge-induced, and a more substantial surface-bulk coupling accelerates both.

The charge-induced degradation, defined by $P_i$, is plotted against the $d_{e\text{-}h}/r_c$ values in Fig. 4g for Q1-Q7. It is worth mentioning that the variation trend of $P_i$ with QDs is consistent with that of EQE because both physical quantities eliminate the wavelength dependence. The results of Q1-Q5 are extracted from Figs. 3 and 4. Q6 and Q7, as the control samples, have the same core as Q4 and Q5, respectively. However, their intermediate shells are thinner than the optimal ones (see $J$-$V$ in Supplementary Fig. 25). As summarized in Fig. 4g, although all the enclosed QDs share the same outer shell and surface configuration, those with larger $d_{e\text{-}h}/r_c$ values generally show faster spectral redshift and charge-induced QD degradation. Conversely, QDs with smaller $d_{e\text{-}h}/r_c$ values are more stable in terms of $P_i$. The effect of reducing surface-bulk coupling is pronounced.

Finally, we discuss the relationship between charge accumulation and surface-bulk coupling. Charge accumulation and surface fixed charges cannot be avoided even in state-of-the-art red QLEDs.

However, the negligible surface-bulk coupling of red QDs shields the excitonic transition from surface localized charges (see Supplementary Fig. 6c and Supplementary Fig. 13), resulting in ultrastable EL. Blue QDs are more prone to charge accumulation than their red counterparts, but that alone does not explain the gap in operational lifetimes.

The seven types of blue QDs summarized in Fig. 4g show optical bandgaps differing by <0.1 eV (see Supplementary Fig. 16i). Moreover, they share the same outer shell and surface configuration, none of which is thick enough to obstruct charge injection. $J-V$ and EQE-$J$ curves also indicate similar charge balance (see Fig. 3b and Supplementary Fig. 26). Therefore, surface-bulk coupling ($d_{e-h}/r_c$) is the determining factor of charge-induced degradation.

In conclusion, although Cd-based QDs with monotonically graded core/shell/shell structures feature facilitated charge injection, near-unity PLQE, and competitive EL stability, our EL-PL analysis proves that charge-induced QD degradation still grows irreversibly during QLED operation. As further revealed by the multiple spectral studies, surface localized charges initiate the irreversible QD degradation, and their impact on excitonic transition is magnified by the abnormal dipolar nature of blue QDs. Inspired by these observations, we adopt two synthetic strategies to reduce the surface-bulk coupling. The increase in core size results in more confined exciton wavefunction. With CIE

**Table 1 | Electroluminescence performance of blue QDs with different surface-bulk coupling**

| QDs | $d_{e-h}/r_c$ | $\lambda$ (nm) | CIE (x,y) | Max. EQE (%) | $T_{95}$ (h, $L_0$ = 1000 cd m$^{-2}$) |
|---|---|---|---|---|---|
| Q1 | 0.617 | 474 | (0.121, 0.095) | 19.2 | 47 |
| Q2 | 0.403 | 478 | (0.112, 0.125) | 19.7 | 151 |
| Q3 | 0.314 | 482 | (0.098, 0.177) | 20.4 | 227 |
| Q4 | 0.307 | 471 | (0.127, 0.081) | 20.8 | 106 |
| Q5 | 0.361 | 467 | (0.135, 0.063) | 20.1 | 75 |

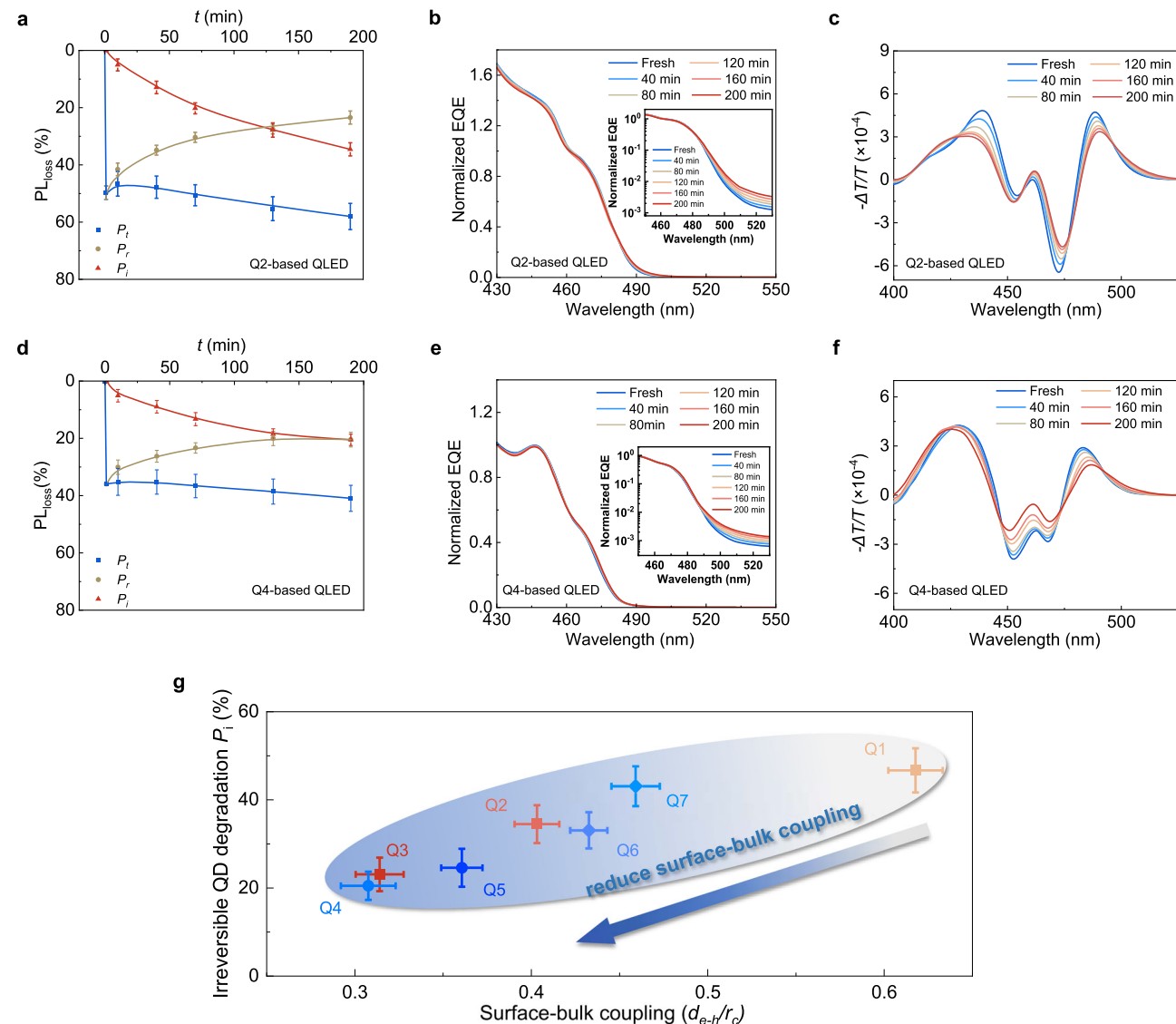

**Fig. 4 | Surface-bulk coupling as a primary degradation mechanism.**
**a**, **d** Reversible, irreversible, and total PL loss of Q2 and Q4 measured from corresponding QLEDs as a function of device operation time. **b**, **e** sEQE$_{PV}$ spectra of Q2- and Q4-based QLEDs. The inset shows the same sEQE$_{PV}$ plotted on a logarithmic scale to demonstrate the near-band-edge transitions. **c**, **f** Electroabsorption spectra of Q2- and Q4-based QLEDs measured as a function of operation time. The devices for time-dependent spectral analysis are aged by a constant current density of 100 mA cm$^{-2}$. **g** The degree of surface-bulk coupling and the rate of charge-induced irreversible degradation of Q1-Q7. As the measure of surface-bulk coupling, the electron-hole separation is normalized to the core size of the nanocrystals ($d_{e-h}/r_c$). The irreversible degradation of QDs is measured using PL-EL. $P_i$ is the irreversible loss of PL photon flux. The error bars represent the data range.

coordinates of (0.112, 0.125) and (0.098, 0.177), QLEDs exhibit record-high $T_{95}$ lifetimes of 151 and 227 h at $L_0 = 1000$ cd m$^{-2}$, respectively. Furthermore, combining the large core and nonmonotonically-graded shell reduces surface-bulk coupling and blueshifts the emission wavelength without compromising charge injection. With CIE coordinates of (0.127, 0.081) and (0.135, 0.063), QLEDs exhibit record-high $T_{95}$ lifetimes of 106 and 75 h at $L_0 = 1000$ cd m$^{-2}$, respectively. When measuring the EL stability by EQE, these QDs are about 30 times higher than the benchmarking device. This work solves the seemingly inherent contradiction between inner-dot surface-bulk coupling and inter-dot charge transport, realizing multiple record-breaking operational lifetimes of QLEDs with emission colors ranging from sky blue to near-Rec.2020-blue.

## Methods

### Chemicals
Zinc acetate (99.99 % trace metals basis), oleic acid (technical grade, 90%), 1-octadecene (ODE, 90%), and Trioctylphosphine (TOP, 90%) were purchased from Sigma-Aldrich. Cadmium oxide (CdO, 99.9%), 1-octadecene (ODE, 90%), selenium powder (Se,100mesh,99.99%), and sulfur powder (S,99.98% trace metals basis) were purchased from Alfa-Aesar. Cadmium oleate is prepared by adding 1 mmol CdO into 1 mL oleic acid and 3 mL ODE, followed by heating to 200 °C for 30 min in Ar flow.

### Synthesis of Q1, Q2, and Q3 blue-QDs: ZnCdSe/ZnCdSeS/ZnS
Zinc acetate (4 mmol for Q1 and 6 mmol for Q2 and Q3) and oleic acid (7 mL) were added to a 50 mL flask and heated to 170 °C with argon flow. Next, 15 mL of 1-octadecene was added to the flask, followed by heating at 300 °C. Then, a total of 0.5 mmol (0.8 mmol for Q2 and 1.5 mmol for Q3) of Se (dissolved in 0.4 mL TOP) mixed with 0.1 mmol (0.16 mmol for Q2 and Q3) Cadmium oleate (dissolved in 0.6 mL oleic acid) was injected into the flask and reacted for 10 min (15 min for Q2 and 25 min for Q3). Next, 0.8 mmol sulfur (S) dissolved in 0.4 mL TOP was injected into the flask and reacted for 5 min. Then, 0.3 mmol cadmium oleate (dissolved in 1.5 mL oleic acid) was injected into the flask and reacted for 30 min. Then, 0.8 mmol S dissolved in 0.4 mL TOP was injected into the flask and reacted for 30 min. The solution was then cooled to room temperature, washed with acetone several times, and finally dissolved in octane to complete the synthesis of ZnCdSe/ZnCdSeS/ZnS QDs.

### Synthesis of Q4 and Q5 blue-QDs: ZnCdSe/ZnSeS/ZnCdS/ZnS
Zinc acetate (6 mmol for Q4 and 9 mmol for Q5) and oleic acid (7 mL) were added into a 50 mL flask and heated to 170 °C with argon flow. Next, 15 mL of 1-octadecene was added to the flask, followed by the rise of temperature to 300 °C. 0.8 mmol of Se (dissolved in 0.4 mL TOP) mixed with 0.16 mmol Cadmium oleate (dissolved in 0.6 mL oleic acid) was injected into the flask (loaded into a 3 mL syringe and injected in a second) and reacted for 15 min. Then, 0.8 mmol Se (dissolved in 0.4 mL TOP) and 0.2 mmol S (dissolved in 0.1 mL TOP) were injected into the flask (0.025 mL min$^{-1}$) and reacted for 30 min. Next, 0.8 mmol S dissolved in 0.4 mL TOP and 0.32 mmol Cd dissolved in 2 mL oleic acid were injected into the flask (0.04 mL min$^{-1}$) simultaneously. Then, 0.8 mmol S dissolved in 0.4 mL TOP was injected into the flask (0.04 mL min$^{-1}$) and reacted for 30 min. The solution was then cooled down to room temperature, subjected to cleaning with acetone several times, and finally dissolved in octane to complete the synthesis of 471 and 467 nm blue-QDs: ZnCdSe/ZnSeS/ZnCdS/ZnS.

### Synthesis of ZnMgO nanoparticles
For a typical synthesis of ZnMgO nanoparticles, 2.8 mmol zinc acetate dihydrate and 0.2 mmol magnesium acetate tetrahydrate were dissolved in 30 mL dimethyl sulfoxide (DMSO). Subsequently, a solution of tetramethylammonium hydroxide (TMAH) dissolved in ethanol (0.6 M) was added to the precursor, followed by stirring for an hour.

Then the solution was centrifuged and washed twice with methanol. The precipitates were finally dispersed in ethanol, forming the ZMO NPs solution with a concentration of 30 mg mL$^{-1}$.

### Characterization of QDs
A FEI Tecnai G2 F20 high-resolution transmission electron microscope (TEM) and an Agilent 5110 inductively coupled plasma optical emission spectroscopy (ICP–OES) were used to study their morphology and elemental compositions. QD samples grown to the different stages were prepared for the radial element distribution. For the absorption measurements, the QDs were spin-coated on quartz, and optical absorbance spectra were acquired with a Shimadzu 3600 UV–VIS-NIR spectrophotometer. The photoluminescence spectra were obtained by using an Edinburgh Instruments FLS1000 fluorescence spectrometer. XRD patterns were recorded by a D8 Advance (Bruker) instrument using a Cu Kα source.

**Device fabrication.** The QLEDs were fabricated by spin coating on glass substrates that were precoated with a patterned indium–tin oxide anode (sheet resistance∼50 Ω per square). The substrates were sequentially cleaned by a tergitol/deionized (DI)-water solution, DI-water, acetone, and isopropanol, respectively, followed by further treatment in a UV ozone cleaner for 15 min. The PEDOT:PSS (Baytron AI 4083) was spin-coated on cleaned substrates at 5000 r.p.m. for 60 s and then baked at 150 °C for 15 min in air. The cooled substrates were then transferred into a nitrogen-filled glove box ($O_2 < 0.01$ ppm, $H_2O < 0.01$ ppm) for subsequent processing. A TFB (American Dye Source) layer was deposited onto the PEDOT:PSS layer by spin-coating its chlorobenzene solution (8 mg mL$^{-1}$, 3000 r.p.m.), followed by baking at 150 °C for 30 min. The quantum dot (in octane) and ZnMgO nanoparticles (in ethanol, ∼30 mg mL$^{-1}$) layers were then spin-coated sequentially onto the substrates and baked at 80 °C for 30 min. With a concentration of 35 mg mL$^{-1}$, the optimized thicknesses of emission layers were achieved by setting the spin speed at 2500 r.p.m. for the red device and 2000 r.p.m for all blue devices. The thickness of the ZnMgO is controlled between 30 and 65 nm by changing the spin speed. Next, the devices were deposited by an Ag cathode through a shadow mask to give a device an active area of 0.04 cm$^2$ under a high vacuum ($<1 \times 10^{-7}$ Torr). Eventually, all devices were encapsulated in a glove box using ultraviolet-curing epoxy and cover glass.

### Device characterization and instrumentation
**Sensitive external quantum efficiency in photovoltaic mode (sEQE$_{PV}$).** Measurements were achieved via a home-built programmed system comprising a 500 W Xenon Arc Lamp (Zolix X500A), a monochromator (Newport CS260), a filter wheel (Newport), an optical chopper (ThorLabs), a lock-in amplifier (Standard Research SR830), a current amplifier (Standard Research SR570), and a calibrated silicon detector (ThorLabs). In the measurement of source spectra, the light from a broadband source passed through a monochromator and then generated a monochromatic beam. Longpass filters with stopband transmission of less than 0.01% were used to eliminate the higher-order diffraction from the monochromator. The monochromatic parallel beam was further modulated by an optical chopper, of which the frequency was used as the reference for a lock-in amplifier (SR 830). The output beam was collected using a Si photodiode connected with a current amplifier (SR 570) and SR 830 in series. In the measurement of the photovoltaic response, the beam spot used for sEQE$_{PV}$ measurement was focused and collimated with a set of optical lenses to reach a spot size less than the sample area. The wavelength-dependent photocurrent was also filtered using an SR 570 and an SR 830.

**Electroabsorption.** A sinusoidal bias with a frequency of 1000 Hz was superimposed on a DC bias of −3 V to modulate the internal electric

field of QLEDs. The negative DC bias also avoided carrier injection and the resulting charge modulation signals. A monochromatic parallel beam produced from the monochromator enters the sample through the ITO side with an incident angle of 45° and is reflected by the back electrode. A calibrated Si photodiode was used to detect the beam reflected from the device. The photodiode was further connected with a current amplifier and a lock-in amplifier. Since the built-in voltage of QLEDs generally increases upon device degradation, the reference frequency was set at the 2nd harmonic frequency to avoid interference from the DC field. As a result, the output is dependent on the square of $V_{ac}$ and independent of $V_{dc}$. The time constant of the lock-in amplifier was set at 1 s, and each data point was averaged from 16 measurements. The final EA signal (-$\Delta T/T$) was the ratio of the signals with and without the modulation of $V_{ac}$. All experiments were carried out at room temperature.

**EL-PL.** For real-time measurements of both EL and PL, a monochromatic source was modulated by a light chopper to excite the QDs in QLEDs. The wavelength of the monochromatic beam was set at 445 nm, which is just below the bandgap of the HTL material. In a typical EL-PL measurement, a Keithley 2400 source meter was used to feed a constant current through the device. The total emission intensity, which comprises a time-independent EL component and a time-dependent (modulated) PL component, was collected by a Si photodiode. The signal further went through a current amplifier and a lock-in amplifier, of which the reference frequency was set as the modulation frequency of the excitation source. The PL component was filtered out by the lock-in amplifier, and the remaining part was identified as the EL intensity. The power density of the excitation source (0.5 mW cm$^{-2}$) was sufficiently low to avoid exciton-exciton quenching.

***J–L–V characteristics and operational lifetime.*** The programmed $J–L–V$ characteristics measurement system is composed of a Keithley 2400 source meter for $J(V)$, a calibrated silicon detector mounted into an integration sphere and further connected with a Keithley 6485 picoammeter for the collection of photons, and an Ocean Optics FLAME-T-VIS-NIR-ES spectrometer for electroluminescence spectra for the calculation of $L(V)$. Luminance was further calibrated by a luminance meter (Konica Minolta LS-160). For the lifetime test, the encapsulated samples were measured under ambient conditions using a commercialized lifetime test system (Guangzhou New Vision Optoelectronic Technology Co. Ltd.).

## Data availability
The data that support the findings of this study are available within the article and its Supplementary Information. All other relevant data are available from the corresponding authors upon request.

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

## Acknowledgements

The work is supported by the National Key Research and Development Program of China (S.C., Grant 2021YFB3601700), the Natural Science Foundation of China (S.C., Grant 61804102), the Pearl River Talent Recruitment Program for Guangdong Introducing Innovative and Enterpreneurial Teams (L.W., 2016ZT06C650), Priority Academic Program Development (PAPD) of Jiangsu Higher Education Institutions (S.C.), and Jiangsu Shuangchuang Plan (S.C.).

## Author contributions

X.C. and M.C. conducted the measurements of EL-PL, sEQE$_{PV}$, and electroabsorption. X.L., X.S., X.C., and R.L. contributed to device fabrication. L.Z., Y.Y., and W.H. contributed to material synthesis. S.C. and L.W. conceived the idea and supervised the experiments. X.Y., X.Z., and W.C. supervised the QLED research at TCL. S.C. wrote the first draft of the manuscript. All authors provided suggestions for revision.

## Competing interests

The authors declare no competing interests.
