## [Peer Review File · Nature Communications]

Blue Light-Emitting Diodes Based on Colloidal Quantum Dots with Reduced Surface-Bulk CouplingEditorial Note: This manuscript has been previously reviewed at another journal that is not operating a transparent peer review scheme. This document only contains reviewer comments and rebuttal letters for versions considered at *Nature Communications*.

REVIEWERS' COMMENTS

Reviewer #2 (Remarks to the Author):

To make the paper better, I suggest the authors to address the below issues before publication.

Authors claimed that the intrinsic stability of blue QDs is not an issue and showed the PL stability of the QDs excited by a source with a power of 2 mW/cm². I am afraid that the excitation power is too weak to age the QDs. Please measure the absorption of the QDs and calculate the effective power density or exciton density in the QDs. To make a fair evaluation, the exciton density of the QDs should be similar with that in QLEDs operated at a relevant brightness.

I cannot find the Supplementary Figure 7e in the revised file.

Reviewer #3 (Remarks to the Author):

The current version of the ms is a worthy contribution to NC.

Reviewer #2 (Remarks to the Author):

To make the paper better, I suggest the authors to address the below issues before publication.

Authors claimed that the intrinsic stability of blue QDs is not an issue and showed the PL stability of the QDs excited by a source with a power of 2 mW/cm². I am afraid that the excitation power is too weak to age the QDs. Please measure the absorption of the QDs and calculate the effective power density or exciton density in the QDs. To make a fair evaluation, the exciton density of the QDs should be similar with that in QLEDs operated at a relevant brightness.

Response: We appreciate the comment.

We set the excitation wavelength at 445 nm (photon energy= 4.47×10^{-19} J), which is just below the bandgap of the HTL material. As shown in Figure below, the devices were measured in reflection mode in which a 100 nm Ag is the reflective mirror. Supplementary Fig.9e (above) shows that Q4 shows no PL degradation during a continuous test of 40 h.

For a typical measurement, the absorbed power of a Q4 layer is:

$$I_{\text{abs}} = 0.1 \text{ mW} / 0.04 \text{ cm}^2 = 2.5 \text{ mW cm}^{-2}$$

Hence, the photon absorption rate by a QD layer is:

$$N_{\text{photon}} = 10^{-4} \text{ J s}^{-1} / 4.47 \times 10^{-19} \text{ J} = 2.23 \times 10^{14} \text{ s}^{-1}$$

Assuming a near-unity PLQE, the volume density excitons ($n_{\text{exciton_PL}}$) and emitted photons ($n_{\text{photon_PL}}$) and can be estimated:

$$n_{\text{exciton_PL}} \approx n_{\text{photon_PL}} \approx 2.23 \times 10^{14} \text{ s}^{-1} / 0.04 \text{ cm}^2 / 20 \text{ nm} \approx 2.8 \times 10^{21} \text{ cm}^{-3} \text{ s}^{-1}$$

We also calculated the exciton density of QDs in QLEDs operated at 200 cd m⁻², which is typical for displays. As shown in Figure 3a, the current density of QLEDs operated at 200 cd m⁻² is about 1.5 mA cm⁻², which translates to electron injection rate:

$$N_e = 1.5 \text{ mA cm}^{-2} \times 0.04 \text{ cm}^2 \times 10^{-3} \text{ C s}^{-1} / (1.6 \times 10^{-19} \text{ C}) = 3.75 \times 10^{14} \text{ s}^{-1}$$

Hence, the number of photons emitted can be calculated using an estimated internal EL quantum efficiency ($\approx 60\%$) at 200 cd m⁻²:

$$N_{\text{photon}} = N_e \times 60\% = 2.25 \times 10^{14} \text{ s}^{-1}$$

Again, assuming a near-unity PLQE, the exciton density is estimated as:

$$n_{\text{exciton_EL}} \approx N_p / 0.04 \text{ cm}^2 / 20 \text{ nm} \approx 2.8 \times 10^{21} \text{ cm}^{-3} \text{ s}^{-1}$$

In terms of exciton generation, the EL-PL measurement condition ($n_{\text{exciton_PL}}$) is roughly the same as a EL emission ($n_{\text{exciton_EL}}$) at 200 cd m^{-2} , which is a typical value for display application. Therefore, the incident power used in our EL-PL is relevant to QLED, suggesting decent intrinsic stability of blue QDs.

I cannot find the Supplementary Figure 7e in the revised file.

Response: Unfortunately, this manuscript has yet to attract this reviewer adequately, but Supplementary Figure 7e has been there since we responded to the reviewer's comments on NPHOT-2022-04-00504a-z. The figure is now labeled Supplementary Figure 9e after the final revision.

Supplementary Fig. 7 | The stability of photoluminescence, a and b, Shelf stability of Q1-Q5; **c and d,** thermal stability of Q1-Q5. The PL spectra were taken by were measured when QD was continuously heated at 90 °C. These QDs show high intrinsic stabilities upon storage over 270 days and continuous heating by 12 h. **e,** Q4's photoluminescence under continuous excitation (2 mW cm^{-2}), indicating ultrahigh intrinsic stability.

Reviewer #3 (Remarks to the Author):

The current version of the ms is a worthy contribution to NC.

Response: We appreciate the comment